# XPRESSyourself: Enhancing, standardizing, and automating ribosome profiling computational analyses yields improved insight into data

**Jordan A. Berg**[1]*, **Jonathan R. Belyeu**[2], **Jeffrey T. Morgan**[1], **Yeyun Ouyang**[1], **Alex J. Bott**[1], **Aaron R. Quinlan**[2,3,4], **Jason Gertz**[5], **Jared Rutter**[1,6]*

1 Department of Biochemistry, University of Utah, Salt Lake City, Utah, United States of America,
2 Department of Human Genetics, University of Utah, Salt Lake City, Utah, United States of America,
3 USTAR Center for Genetic Discovery, University of Utah, Salt Lake City, Utah, United States of America,
4 Department of Biomedical Informatics, University of Utah, Salt Lake City, Utah, United States of America,
5 Department of Oncological Sciences, University of Utah, Salt Lake City, Utah, United States of America,
6 Howard Hughes Medical Institute, University of Utah, Salt Lake City, Utah, United States of America

* jordan.berg@biochem.utah.edu (JAB); rutter@biochem.utah.edu (JR)

**Data Availability Statement:** The data used in this manuscript are available under the Gene Expression Omnibus persistent identifier

## Abstract

Ribosome profiling, an application of nucleic acid sequencing for monitoring ribosome activity, has revolutionized our understanding of protein translation dynamics. This technique has been available for a decade, yet the current state and standardization of publicly available computational tools for these data is bleak. We introduce XPRESSyourself, an analytical toolkit that eliminates barriers and bottlenecks associated with this specialized data type by filling gaps in the computational toolset for both experts and non-experts of ribosome profiling. XPRESSyourself automates and standardizes analysis procedures, decreasing time-to-discovery and increasing reproducibility. This toolkit acts as a reference implementation of current best practices in ribosome profiling analysis. We demonstrate this toolkit's performance on publicly available ribosome profiling data by rapidly identifying hypothetical mechanisms related to neurodegenerative phenotypes and neuroprotective mechanisms of the small-molecule ISRIB during acute cellular stress. XPRESSyourself brings robust, rapid analysis of ribosome-profiling data to a broad and ever-expanding audience and will lead to more reproducible and accessible measurements of translation regulation. XPRESSyourself software is perpetually open-source under the GPL-3.0 license and is hosted at https://github.com/XPRESSyourself, where users can access additional documentation and report software issues.

This is a *PLOS Computational Biology* Software paper.

GSE65778 for ribosome profiling data and under the dbGaP Study Accession persistent identifier phs000178 for the TCGA data. All scripts used in the analysis of these data are available at https://github.com/XPRESSyourself/xpressyourself_manuscript.

**Funding:** J.A.B. received support from the National Institute of Diabetes and Digestive and Kidney Diseases (NIDDK) Inter-disciplinary Training Grant T32 Program in Computational Approaches to Diabetes and Metabolism Research, 1T32DK11096601 to Wendy W. Chapman and Simon J. Fisher (https://www.niddk.nih.gov/). J.T.M. received support as an HHMI Fellow of the Jane Coffin Childs Memorial Fund for Medical Research (https://www.jccfund.org/). A.J.B received support from the National Cancer Institute (NCI) Predoctoral to Postdoctoral Fellow Transition Award, K00CA212445 (https://www.cancer.gov/). This work was supported by NIDDK fellowship 1T32DK11096601 (to J.A.B.) (https://www.niddk.nih.gov/) and NIH grant R35GM13185 (to J.R.) (https://www.nih.gov/). The computational resources used were partially funded by the NIH Shared Instrumentation Grant 1S10OD021644-01A1 (https://www.nih.gov/). The funders had no role in study design, data collection and analysis, decision to publish, or preparation of the manuscript.

**Competing interests:** The authors have declared that no competing interests exist.

## Introduction

High-throughput sequencing data has revolutionized biomedical and biological research. One such application of this consequential technology is ribosome profiling, which, coupled with bulk RNA-Seq, measures translation efficiency, translation pausing, novel protein translation products, and more [1–3]. Though the experimental procedures for ribosome profiling have matured, an abundance of biases and peculiarities associated with each analytical method or tool are still present and may often be obscured to a new user of this methodology [4–8]. Additionally, standardized methods for handling this unique data type remain elusive. This has been problematic and evidenced by various studies using vague or opaque methods for data analysis (for examples, see [9–13]), or methods rely on outdated tools [5]. Very few labs have the tools necessary to separate the biological signals in ribosome profiling data from the inherent biases of the experimental measurements, and these tools are not readily accessible by the community. This is a critical time in the rapidly expanding influence of ribosome profiling. For too long, the bioinformatic know-how of this incredibly powerful technique has been limited to a small handful of labs. As more and more ribosome profiling studies are performed, more and more labs will lack the ability to analyze their data with ease and fidelity. Few, if any, extant pipelines or toolkits offer a thorough set of integrated tools for assessing standard quality control metrics or performing proper reference curation to reduce systematic biases across any organism, particularly with ribosome profiling data [14–18].

For example, one issue in ribosome profiling is the pile-up of ribosomes at the 5′- and 3′-ends of coding regions within a transcript, a systematic biological signal arising from the slower kinetics of ribosome initiation and termination compared to translation elongation and is generally regarded to not accurately reflect measurements of translation efficiency. These signals are further exacerbated by pre-treatment with cycloheximide during ribosome footprint harvesting [4, 19, 20]. These pile-ups can dramatically skew ribosome footprint quantification and measurements of translational efficiency. Current practices in the field recommend excluding pile-up-prone regions when quantifying ribosome profiling alignments as they lead to noisier estimations of translation efficiency [3, 21]; however, no publicly available computational tools currently exist to facilitate these automated adjustments to reference transcripts. Curating references properly and robustly requires advanced implementations. In addition, downstream data visualization methods presently available are often not optimized to analyze and compare translation regulatory regions of a gene.

To address deficiencies in the public ribosome profiling computational toolkit, we developed XPRESSyourself, a computational toolkit and adaptable, end-to-end pipeline that bridges these and other gaps in ribosome profiling data analysis. XPRESSyourself implements the complete suite of tools necessary for comprehensive ribosome profiling and bulk RNA-Seq data processing and analysis in a robust and easy-to-use fashion, often packaging tasks that would typically require hundreds to thousands of lines of code into a single command. For instance, XPRESSyourself creates the mRNA annotation files necessary to remove confounding systematic factors during quantification and analysis of ribosome profiling data, allowing for accurate measurements of translation efficiency. It provides the built-in capacity to quantify and visualize differential upstream open-reading frame (uORF) usage by generating IGV-like, intron-less plots for easier visualization [22]. The ability to visualize (and in another XPRESSyourself module, quantify) the usage of micro-uORFs is important in exploring regulatory events or mechanisms in a wide array of biological responses and diseases. XPRESSyourself also introduces a tool for efficient identification of the most problematic rRNA (ribosomal RNA) fragments for targeted depletion, which provides immense financial and experimental benefits to the user by amplifying ribosome footprint signal over rRNA noise.

Tools like this will become vital as ribosome profiling moves into development in new organisms.

XPRESSyourself aims to address the lack of consensus in analytical approaches used to process ribosome profiling data by acting as a reference implementation of current best practices for ribosome profiling analysis. While a basic bioinformatic understanding is becoming more commonplace amongst the scientific community, the intricacies of processing RNA-Seq data remain challenging for many. Moreover, many users are often not aware of the most up-to-date tools or the appropriate settings for their application [23, 24]. Even for the experienced user, developing robust automated pipelines that accurately process and assess the quality of these datasets can be laborious. The variability that inevitably arises with each lab or core facility designing and using distinct pipelines is also a challenge to reproducibility in the field. XPRESSyourself curates the state-of-the-art methods for use and where a required functionality is unavailable, introduces a thoroughly tested module to fill that gap. While some tasks in these pipelines may be considered mundane, we eliminate the need of each user to rewrite even simple functionality and promote reproducibility between implementations. To aid users of any skill-level in using this toolkit, we provide thorough documentation, walkthrough videos, and interactive command builders to make usage as easy as possible, while allowing for broad use of this toolkit from personal computers to high-performance clusters.

Finally, the most broadly relevant aspect of our update and streamlining of ribosome-profiling analysis is the novel biological insights we are able to obtain from published datasets. We highlight this in the ISRIB ribosome-profiling study discussed in this manuscript, where we are able to observe significant translation regulation that was missed previously when the data were initially analyzed using now outdated techniques. This analysis generates novel hypotheses for genes potentially involved in neurodegeneration in humans, but more broadly emphasizes the benefit of analysis and re-analysis of data using the complete and up-to-date benchmarked methodology provided within XPRESSyourself.

## Design and implementation

### Architecture and organization

XPRESSyourself is currently partitioned into two software packages, XPRESSpipe and XPRESSplot. XPRESSpipe contains automated, end-to-end pipelines tailored for ribosome profiling, single-end RNA-Seq, and paired-end RNA-Seq datasets. Fig 1 outlines the tasks completed by these pipelines. Individual sub-modules can be run automatically through a pipeline or manually step-by-step. Modules optimize available computational resources where appropriate to deliver results as quickly as possible. XPRESSplot is available as a Python library and provides an array of analytical methods specifically for sequence data, but tractable to other data types. For a comparison of how XPRESSyourself compares to other available software packages available at the time of writing, we refer the reader to S1 Fig [14, 15, 18, 25–49].

To make analysis as easy and accessible as possible, an integrated command builder for reference curation and sample analysis can be run by executing `xpresspipe build`. This command builder will walk the user through potential considerations based on their library preparation method and build the appropriate command for execution on their personal computer or a supercomputing cluster. The builder will then output the requested command for use on a computational cluster, or the command can be executed immediately on a personal computer.

The software is designed such that updating and testing of a new module, or updating dependency usage, are facile tasks for a trained bioinformatician. More details on current and future capabilities can be found in each package's documentation [50, 51], their respective

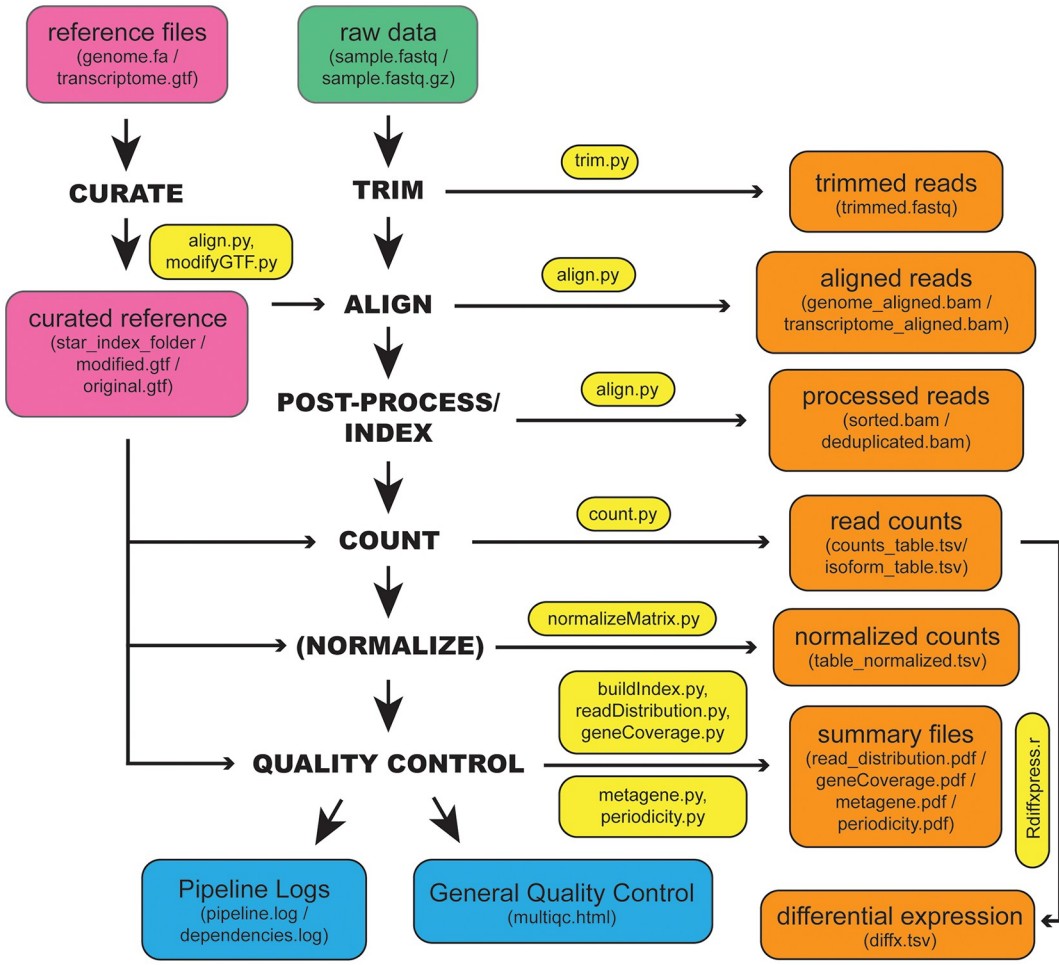

**Fig 1. Workflow schematic of the inputs, outputs, and organization of XPRESSpipe.** Representation of the general steps performed by XPRESSpipe with data and log outputs. Steps in parentheses are optional to the user. Input and output file types are in parentheses for each input or output block. The main script(s) used for a given step are in yellow blocks. The green block indicates input sequence file(s). Pink blocks indicate reference input files and curated reference. Orange blocks indicate output files. Blue blocks indicate general quality control and log file outputs. Differential expression analysis is run independently from the pipeline as the user will need to ensure count table and metadata table formatting are correct before use.

versions page on each toolkit's repository page [52], or via archived supplemental files at time of writing in S2 and S3 Files.

## Automated reference curation

The first step of RNA-Seq alignment is curating an organism reference to which the alignment software will map sequence reads. XPRESSpipe uses STAR [53] for mapping reads as it has been shown consistently to be the best performing RNA-Seq read aligner for the majority of cases [54, 55]. The appropriate reference files are automatically curated by providing the appropriate GTF file saved as transcripts.gtf and the directory path to the genomic FASTA file(s). Additional modifications to the GTF file required for ribosome profiling or desired for RNA-Seq are discussed in the next section. We recommend organizing these files in their own directory per organism.

## GTF modification

For ribosome profiling, frequent read pile-ups are observed at the 5′- and 3′- ends of an open reading frame which are largely uninformative to a gene's translational efficiency [4]. While these pile-ups can be indicative of true translation dynamics [56], current best-practices have more recently settled on ignoring these regions during read quantification and calculations of translation efficiency [3, 21]. By providing the `--truncate` argument during reference curation, the 5′- and 3′- ends of each coding region will be recursively trimmed until the specified amounts are removed from coding space. A recursive strategy is required here as GTF file-formats split the CDS (coding sequence of gene) record into regions separated by introns. By default, 45 nt (nucleotides) will be trimmed from the 5′-ends and 15 nt from the 3′-ends recursively until the full length is removed from coding space, as is the current convention within the ribosome profiling field [3]. The resulting output file will then be used to process ribosome footprint libraries and their corresponding bulk RNA-Seq libraries. If generating a GTF file for use solely with general bulk RNA-Seq datasets, this file should not be truncated.

Optionally, the GTF can be parsed to retain only protein-coding gene records. This acts as a read masking step to exclude non-protein coding transcripts. In particular, overabundant ribosomal RNAs resulting during library preparation are excluded from downstream analyses using this modified reference file. Parameters can also be provided to retain only the Ensembl canonical transcript record. This can be useful for some tools that penalize reads that overlap multiple isoforms of the same gene. If using HTSeq with default XPRESSpipe parameters or Cufflinks to quantify reads, this is not necessary as they do not penalize a read mapping to multiple isoforms of the same gene or are capable of handling quantification of different isoforms of a gene [57, 58].

## Read processing

**Pre-Processing.**   In order for sequence reads to be mapped to the genome, reads generally need to be cleaned of artifacts from library creation. These include adaptors, unique molecular identifier (UMI) sequences, and technical errors in the form of low-quality base calls. Parameters, like minimum acceptable base quality or minimum and maximum read length, can be modified, or features such as unique molecular identifiers (UMIs) can be specified to identify and group PCR artifacts for later removal [59, 60].

**Alignment.**   Reads are aligned to the reference genome with STAR, which, despite being more memory-intensive, is one of the fastest and most accurate sequence alignment options currently available [53–55]. XPRESSpipe is capable of performing a single-pass, splice-aware, GTF-guided alignment or a two-pass alignment of reads wherein novel splice junctions are determined and built into the genome index, followed by alignment of reads using the updated index. Both coordinate- and transcriptome-aligned BAM files are output by STAR. We generally abstain from rRNA negative alignment at this step as downstream analysis of these mapped reads could be of interest to some users; however, the option to remove these alignments is available to users via the appropriate argument. When rRNA quantification is not preferred, a protein-coding-only GTF file should be provided during quantification. A STAR-compatible VCF file can also be passed to this step to allow for genomic variant consideration during alignment.

**Post-processing.**   XPRESSpipe further processes alignment files by sorting, indexing, and optionally parsing unique alignments based on UMIs for downstream analyses. PCR duplicates are also detected and marked or removed for downstream analyses; however, these files are only used for relevant downstream steps or if the user specifies to use these de-duplicated files in all downstream steps. Use of de-duplicated alignment files may be advisable in

situations where the library complexity profiles (discussed below) exhibit high duplication frequencies. However, generally the abundance of PCR-duplicates is low in properly-prepared sequencing libraries; thus, doing so may be overly stringent and unnecessary [59]. Optionally, BED coverage files can also be output.

**Quantification.** XPRESSpipe quantifies read alignments for each input file using HTSeq with the `intersection-nonempty` method by default [57, 61]. We use this quantification method as it conforms to the current TCGA (The Cancer Genome Atlas) processing standards and is favorable in the majority of applications [62]. If masking of non-coding RNAs or quantification to truncated CDS records is desired, a protein-coding-only GTF file should be provided to the `--gtf` argument. HTSeq importantly allows selection of feature type across which to quantify, thus allowing for quantification across the CDSs of a transcript instead of entire exons. If a user is interested in quantifying ribosome occupancy of transcript uORFs in ribosome footprint samples, they can provide `five_prime_utr` or `three_prime_utr` for the `--feature_type` parameter if such annotations exist in the organism of interest's GTF file. If the user is interested in isoform abundance estimation of reads, Cufflinks is alternatively available for quantification [58, 61].

**Normalization.** Methods for count normalization are available within XPRESSpipe by way of the XPRESSplot package. For normalizations correcting for transcript length, the appropriate GTF must be provided. Sample normalization methods available include reads per million (RPM), reads per kilobase of transcript per million (RPKM) or fragments per kilobase of transcript per million (FPKM), and transcripts per million (TPM) normalization [63]. For samples sequenced on different chips, prepared by different individuals, or on different days, the `--batch` argument may be recommended along with the appropriate metadata matrix [64].

## Quality control

**Read length distribution.** The lengths of all reads are analyzed after trimming. By assessing the read distribution of each sample, the user can ensure the expected read size was sequenced. This is particularly helpful in ribosome profiling experiments for verifying the requisite 17-33 nt ribosome footprints were selectively captured during library preparation [3, 65]. Metrics here, as in all other quality control sub-modules, are compiled into summary figures for easy pan-sample assessment by the user.

**Library complexity.** Measuring library complexity is an effective method for analyzing the robustness of a sequencing experiment in capturing various, unique RNA species. As the majority of RNA-Seq preparation methods involve a PCR step, sometimes particular fragments will be favored and over-amplified in contrast to others. By plotting the number of PCR replicates versus expression level for each gene, one can monitor any effects of limited transcript capture diversity and high estimated PCR duplication rate on the robustness of their libraries. This analysis is performed using dupRadar [66] using the duplicate-tagged alignment files output during post-processing. Metrics are then compiled and plotted by XPRESSpipe.

**Metagene estimation profile.** To identify any general biases for the preferential capture of the 5′- or 3′- ends of transcripts, metagene profiles are generated for each sample. This is performed by determining the meta-genomic coordinate for each aligned read in exon space. Coverage is calculated for each transcript, normalized, and combined to eliminate greediness of super-expressors in profile coverage. Required inputs are an indexed BAM file and an unmodified GTF reference file. Outputs include metagene metrics, individual plots, and summary plots. Parameters can be tuned to only analyze representation along CDS regions.

**Gene coverage profile.** Extending the metagene estimation analysis, the user can focus on the coverage profile across a single gene. Although traditional tools like IGV [22] offer the ability to perform such tasks, XPRESSpipe offers the ability to collapse the introns to observe coverage over exon space only. This is helpful in situations where massive introns spread out exons and make it difficult to visualize exon coverage for the entire transcript in a concise manner. CDS feature annotations are displayed to aid ribosome profiling data users in identifying CDS coverage and uORF translation events. When running the XPRESSpipe pipeline, a housekeeping gene will be automatically processed and output for the user's reference. S2 Fig provides a comparison with the output of IGV [22] and XPRESSpipe's `geneCoverage` module over a similar region for two genes to demonstrate the compatibility between the methods. We note that while the published superTranscripts tool offers similar functionality, it lacks integration and automation and must be manually paired with IGV for multi-sample comparisons and visualization [31]. Other tools, such as Rfeet and riboStreamR [25, 47], suffer from similar integration and automation shortcomings. XPRESSpipe's `geneCoverage` module offers easy and automated functionality for this task.

**Codon phasing/periodicity estimation profile.** In ribosome profiling, a useful measure of a successful experiment is obtained by investigating the codon phasing of ribosome footprints [3]. To do so, the P-site positions relative to the start codon of each mapped ribosome footprint are calculated using riboWaltz [67], along with the codon usage distribution. The same inputs are required as in the `metagene` sub-module.

**Identifying problematic rRNA fragments from ribosome footprinting for depletion.** rRNA depletion is intrinsically complicated during the preparation of ribosome-footprint profiling libraries: poly(A) selection is irrelevant, and kit-based rRNA depletion is grossly insufficient. Especially in the case of ribosome profiling experiments, where RNA is digested by an RNase to create ribosome footprints, many commercial depletion kits will not target the most abundant rRNA fragment species produced during the footprinting step of ribosome profiling. The sequencing of these RNAs becomes highly repetitive, wasteful, and typically biologically uninteresting in the context of gene expression and translation efficiency. The depletion of these sequences is therefore desired to increase the depth of coverage of ribosome footprints. Depending on the species and condition being profiled, custom rRNA-depletion probes for a small subset of rRNA fragments (generally 2-5) can easily account for more than 90% of sequenced reads [1, 3]. The `rrnaProbe` sub-module analyzes the over-represented sequences within a collection of footprint sequence files that have already undergone adaptor and quality trimming, compiles conserved sequences across the overall experiment, and outputs a rank-ordered list of these sequences for probe design. The user should then determine which of these most abundant sequences map to rRNAs.

## Analysis

XPRESSpipe provides a DESeq2 command-line wrapper for performing differential expression analysis of count data. We refer users to the original publication for more information about uses and methodology [68].

More analytical features are available in XPRESSplot, which requires as input a gene count table as output by XPRESSpipe and a meta-sample table (explained in the documentation [51], or see S3 File). Analyses with limited to no options in Python libraries include principle components plotting with confidence intervals and automated volcano plot creation for RNA-Seq or other data. Other instances of analyses can be found in the documentation [51] (or see S3 File).

## Results

### Benchmarking against published ribosome profiling data and new insights

The integrated stress response (ISR) is a signaling mechanism used by cells and organisms in response to a variety of cellular stresses [69]. Although acute ISR activation is essential for cells to properly respond to stresses, long periods of sustained ISR activity can be damaging. These prolonged episodes contribute to a variety of diseases, including many resulting in neurological decline [70]. A recently discovered small-molecule inhibitor of the ISR, ISRIB, has been demonstrated to be a potentially safe and effective neuroprotective therapeutic for traumatic brain injury and other neurological diseases. Interestingly, ISRIB can suppress the damaging chronic low activation of the ISR, while it does not interfere with a cytoprotective acute, high-grade ISR, adding to its wide pharmacological interest [9, 71–76].

A recent study (data available under Gene Expression Omnibus accession number GSE65778) utilized ribosome profiling to better define the mechanisms of ISRIB action on the ISR, modeled by 1-hour tunicamycin (Tm) treatment in HEK293T cells [9]. A key finding of this study is that a specific subset of stress-related transcription factor mRNAs exhibits increased translational efficiency (TE) compared to untreated cells during the tunicamycin-induced ISR. However, when cells were co-treated with tunicamycin and ISRIB, the TE of these stress-related mRNAs showed no significant increase compared to untreated cells, which indicates that ISRIB can counteract the translational regulation associated with the ISR.

To showcase the utility of XPRESSpipe in analyzing ribosome profiling and sequencing datasets, we re-processed and analyzed this dataset using the more current *in silico* techniques included in the XPRESSpipe package to further query the translational mechanisms of the ISR and ISRIB. All XPRESSpipe-processed biological replicate samples exhibited a strong correlation between read counts per gene when thresholded similarly to count data available with the original publication (Spearman $\rho$ values 0.991-0.997) (Fig 2A shows representative plots; Pane A in S3 Fig shows all replicate comparisons; Pane B in S4 Fig shows the corresponding plots using the count data provided with the original publication for reference).

Compared to the count data made available with the original manuscript, when XPRESSpipe-processed samples were thresholded as in the original published count data, samples showed generally comparable read counts per gene between the two analytical regimes (Spearman $\rho$ values 0.937-0.951) (Fig 2B shows representative plots; S3B Fig shows all comparisons). This is in spite of the fact that the methods section of the original publication employed software that was current at the time but is now outdated, such as TopHat2 [77], which has a documented higher false-positive alignment rate, generally lower recall, and lower precision at correctly aligning multi-mapping reads compared to STAR [53–55]. Many of the genes overrepresented in the original count data as compared to data processed by XPRESSpipe appear to be due to the over-estimation of pseudogenes or other gene paralogs. Pane A in S4 Fig highlights a sampling of some extreme cases where particular genes with paralogs are consistently over-represented between samples in the original processed data. This suggests a programmatic difference in how these transcripts are being treated. As these genes share high sequence similarity with each other, reads mapping to these regions are difficult to attribute to a specific genomic locus and are often excluded from further analyses due to their multi-mapping nature. The benchmarking study [54] that examined these and other aligners described how TopHat2 had a disproportionally high rate of incorrectly aligned bases or bases that were aligned uniquely when they should have been aligned ambiguously, at least partially explaining the observed overcounting effect with TopHat2. Had TopHat2 marked problematic reads as ambiguous, they would have been excluded from later quantification.

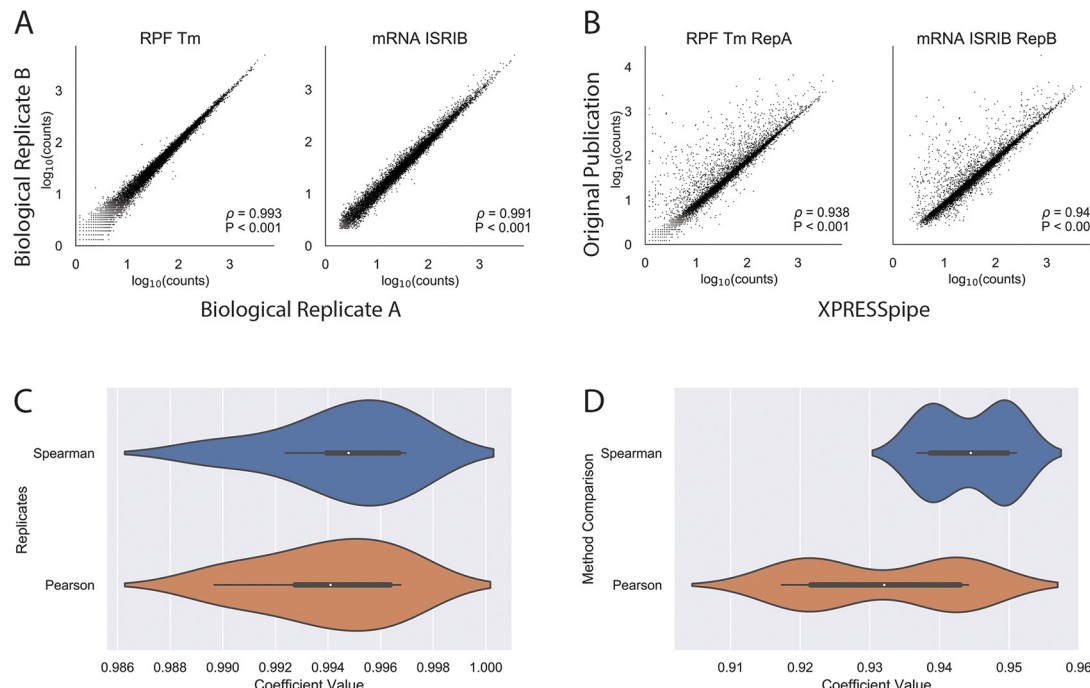

**Fig 2. Representative comparisons between processed data produced by XPRESSpipe and original study.** Genes were eliminated from analysis if any RNA-Seq sample for that gene had fewer than 10 counts. A) Representative comparisons of biological replicate read counts processed by XPRESSpipe. B) Representative comparisons of read counts per gene between count data from the original study and the same raw data processed and quantified by XPRESSpipe. C) Boxplot summaries of Spearman $\rho$ and Pearson r values for biological replicate comparisons. D) Boxplot summaries of Spearman $\rho$ and Pearson r values for between method processing. RPF, ribosome-protected fragments. Tm, tunicamycin. All $\rho$ values reported in A and B are Spearman correlation coefficients using RPM-normalized count data. Pearson correlation coefficients were calculated using $\log_{10}(\text{rpm}(\text{counts}) + 1)$ transformed data. XPRESSpipe-processed read alignments were quantified to *Homo sapiens* build CRCh38v98 using a protein-coding-only, truncated GTF.

Additionally, when TopHat2 and STAR were tested using multi-mapper simulated test data of varying complexity, TopHat2 consistently suffered in precision and recall. These calls are increasingly more difficult to make with smaller reads as well, and this is evident from Fig 2B, where ribosome footprint samples consistently showed more over-counted reads than the corresponding RNA-Seq samples. When dealing with a ribosome footprint library of about 50-100 million reads, and with TopHat2's simulated likelihood of not marking an ambiguous read as such being about 0.5% higher than STAR, this would lead to around 250,000 to 500,000 spuriously aligned reads, which is in line with our observations (statistics were derived from [54]; analyses are available in the manuscript scripts repository [78]).

Another potential contributor to this divergence is that the alignment and quantification within XPRESSpipe use a current human transcriptome reference, which no doubt contains updates and modifications to annotated canonical transcripts and so forth when compared to the version used in the original study. However, in practice, these effects are modest for this dataset (S5 Fig). Additionally, the usage of the now outdated DESeq1 [79] appears to contribute significantly to the outcome in differential expression analysis (S6 Fig). While differences in processing between the outdated and current methods may not always create systematic differences in output, key biological insights may be missed. The analysis that follows is exploratory and only meant to suggest putative targets identifiable by re-analyzing pre-existing, publicly available data.

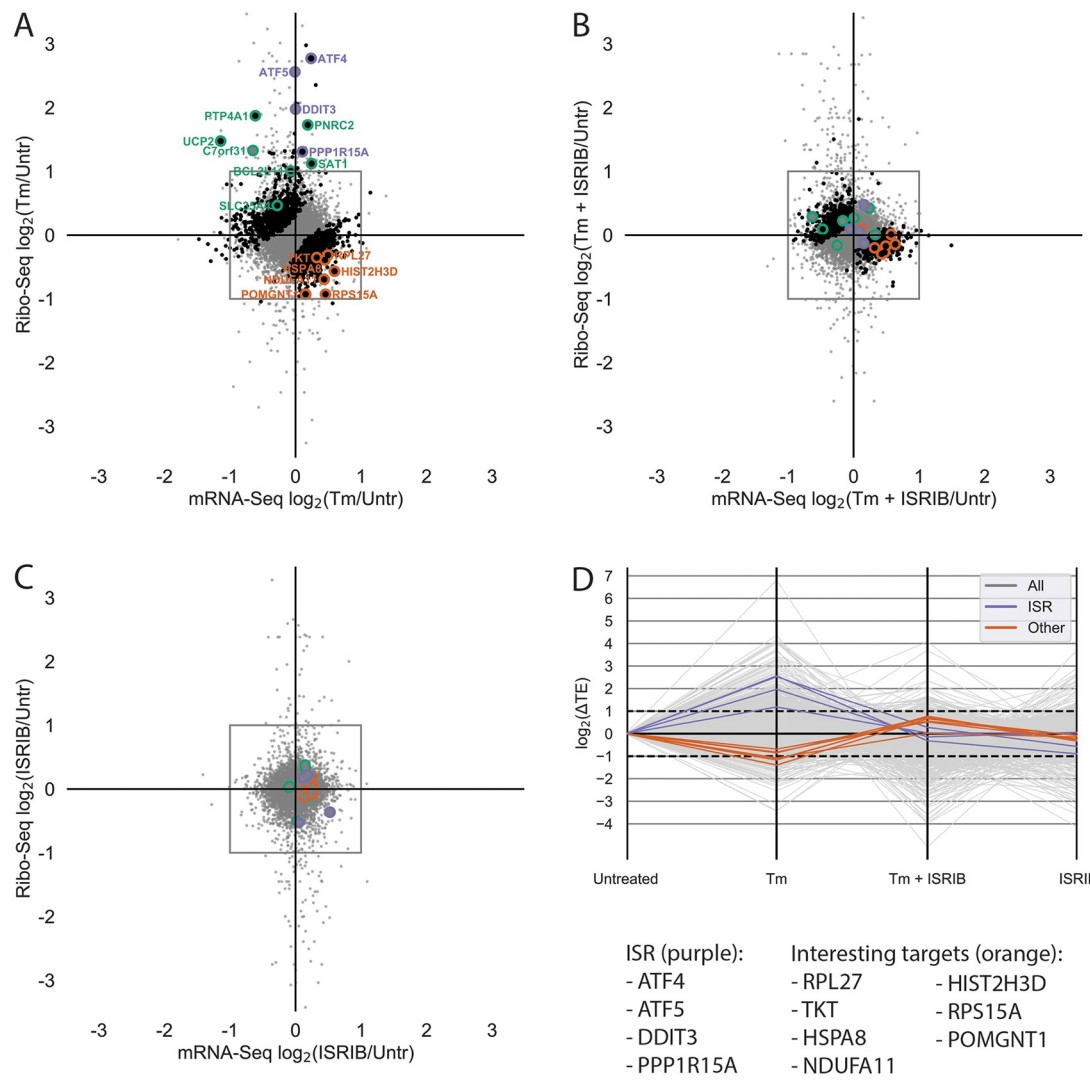

**Fig 3. Analysis of previously published ISR TE data using XPRESSpipe.** A-C) log₂(Fold Change) for each drug condition compared to untreated for the Ribo-seq and RNA-Seq data. Purple, ISR canonical targets highlighted in the original study. Green, genes with uORFs affected by ISR as highlighted in the original study. Orange, genes fitting a strict TE thresholding paradigm to identify genes that display a 2-fold or greater increase in TE in Tm + ISRIB treatment compared to Tm treatment. Black, genes with statistically significant changes in TE. Grey, all genes. Changes in Ribo-seq and mRNA-Seq were calculated using DESeq2. TE was calculated using DESeq2. Points falling outside of the plotted range are not included. D) Changes in log₂(TE) for each drug condition compared to untreated control. Grey, all genes. Purple, ISR targets identified in the original study. Orange, genes fitting a strict TE thresholding paradigm to identify genes that display a 2-fold or greater increase in TE in Tm + ISRIB treatment compared to Tm treatment. XPRESSpipe-processed read alignments were quantified to *Homo sapiens* build CRCh38v98 using a protein-coding-only, truncated GTF.

We first examined the canonical targets of translation regulation during ISR, as identified in the original study within the XPRESSpipe-processed data. These targets include ATF4, ATF5, PPP1R15A, and DDIT3 (Fig 3A–3C, highlighted in purple) [9]. Of note, the fold-change in ribosome occupancy of ATF4 (6.83) from XPRESSpipe-processed samples closely mirrored the estimate from the original publication (6.44). Other targets highlighted in the original study [9], such as ATF5, PPP1R15A, and DDIT3 also demonstrated comparable increases in their ribosome occupancy fold-changes to the original publication count data (XPRESSpipe: 5.90, 2.47, and 3.94; respectively. Original: 7.50, 2.70, and 3.89; respectively)

(Fig 3A). Similar to the originally processed data, all of these notable changes in ribosome occupancy return to untreated levels during Tm + ISRIB co-treatment (Fig 3B). Additional ISR targets containing micro-ORFs described in the study (highlighted in green in Fig 3A–3C) were also similar in translational and transcriptional regulation across conditions between the two analytical regimes.

Both the original study and our XPRESSpipe-based re-analysis show that ISRIB can counteract the significant increase in TE for a set of genes during ISR. To further build upon the original analysis and explore TE regulation during ISR, we asked if ISRIB has a similar muting effect on genes with significant decreases in TE induced by the ISR. In the original study, genes with significant decreases in TE were reported in a source-data table and not focused on in the study. However, re-analysis of these data with the updated XPRESSpipe methodology identifies genes with apparent translational down-regulation that may play a role in the neurodegenerative effects of ISR and the neuroprotective properties of ISRIB [73–76]. Importantly, several of these genes were not identified as having significantly down-regulated TEs in the original analysis, which suggests a rationale for not focusing on translational downregulation. In all, we identified seven genes with the regulatory paradigm of interest: significant decreases in TE during tunicamycin-induced ISR that are restored in the ISR + ISRIB condition (Table 1, descriptions sourced from [80–82]; Fig 3D). RNA-Seq and ribosome-footprint coverage across these genes show that the significant changes in their TE are due to neither spurious, high-abundance fragments differentially present across libraries nor variance from an especially small number of mapped reads (S7 Fig). This is an important consideration as the commonly suggested use of the CircLigase enzyme in published ribosome profiling library preparation protocols, which circularizes template cDNA before sequencing, can bias certain molecules' incorporation into sequencing libraries based on read-end base content alone [83].

**Table 1. Translationally down-regulated genes during acute Tm treatment with restored regulation during Tm + ISRIB treatment.**

| Gene Name | Relevant Description |
|---|---|
| POMGNT1[&] | Participates in O-mannosyl glycosylation. Mutations have been associated with muscle-eye-brain diseases and congenital muscular dystrophies. Expressed especially in astrocytes, as well as in immature and mature neurons. Expressed across brain. |
| RPL27[*&] | Subunit of ribosome catalyzing protein synthesis. Expressed in cerebral cortex in embryonic tissue and/or stem cells. Mutations associated with Diamond-Blackfan Anemia 16, a metabolic disease, which may present with microcephaly. |
| TKT[*&] | Encodes thiamine-dependent enzyme that channels excess sugars phosphates to glycolysis. Mutations associated with developmental delays and Wernicke-Korsakoff Syndrome, a metabolic and neuronal disease and associated with encephalopathy and dementia-like characteristics. |
| HSPA8[*&] | Encodes heat shock protein 70 member. Facilitates protein folding and localization. Diseases associated with mutations include Auditory System Disease and Brain Ischemia, both neurological disorders. Expressed in cerebral cortex in embryonic tissue and/or stem cells. |
| NDUFA11[&] | Encodes subunit of mitochondrial complex I, a vital component of the electron transport chain. Mutations are associated with severe mitochondrial complex I deficiency. Related pathways include the GABAergic synapse. Associated diseases include brain atrophy, encephalopathy, and leber hereditary optic neuropathy. Overexpressed in frontal cortex. |
| HIST2H3D | Responsible for nucleosome structure. No neurological phenotypes currently annotated. |
| RPS15A[*] | Subunit of ribosome catalyzing protein synthesis. Diseases associated include Diamond-Blackfan Anemia, an inborn error of metabolism disease. |

Gene names succeeded by an asterisk (*) indicate these genes were identified in the original data when re-analyzed with DESeq2 [68]. Gene names succeeded by an ampersand (&) indicate genes with strong neurological phenotype annotations. None of these genes were present in the original analysis tables.

Five (POMGNT1, RPL27, TKT, HSPA8, NDUFA11) out of the seven identified genes have annotated neurological functions or mutations that cause severe neurological disorders. Mutations in one other gene (RPS15A) generally result in metabolic disorders. While none of these genes were identified as being of interest in the original study using the original methods, by re-processing the original manuscript count data with DESeq2 [68] and the same expression pattern thresholding, four of these genes are now present in the analysis (RPL27, TKT, HSPA8, RPS15A) (see S6 Fig for a systematic comparison). NDUFA11 and TKT are protein-coding genes whose functions are integrally tied to successful central carbon metabolism and mitochondrial electron transport chain function, respectively. NDUFA11 encodes a subunit of mitochondrial respiratory complex I [84], and TKT encodes a thiamine-dependent enzyme that channels excess sugars phosphates into glycolysis as part of the pentose phosphate pathway [85]. Mutations in NDUFA11 cause severe neurodegenerative phenotypes such as brain atrophy and encephalopathy [84], and mutations in TKT cause diseases associated with neurological phenotypes [86]. These regulatory and phenotypic observations raise the possibility that their role may be functionally relevant to the neurodegenerative effects of ISR and the neuroprotective properties of ISRIB.

While at this stage speculative, it is interesting that the processing of these data with updated methods provides a very conservative list of differentially expressed genes, and that the majority of those genes are associated with severe neurological phenotypes. It is therefore easy to speculate that TE regulation of these targets' abundance might be important in the neurodegeneration observed in prolonged ISR conditions. ISRIB's neuroprotective effects may stem from a restoration of one or more of these entities' protein expression. Though hypothetical without further validation, these ISRIB-responsive neuronal targets act as interesting cases for further validation and study in a model more representative of neurotoxic injury and disease than the HEK-293T model used in the original study. In all, this comparison demonstrates the utility of XPRESSpipe for rapid, user-friendly analysis and re-analysis of ribosome-profiling experiments in the pursuit of biological insights and hypothesis generation.

## Cost analysis and performance

XPRESSpipe functions can be computationally intensive. Super-computing resources are recommended, especially when handling large datasets or when aligning to larger, more complex genomes. Many universities provide super-computing resources to their affiliates; however, in cases where these resources are not available, servers such as Amazon Web Services (AWS) [87] can be used to process sequencing data using XPRESSpipe. Table 2 summarizes the runtime statistics for the ISRIB dataset used in this study. The ISRIB ribosome profiling dataset contained a total of 32 raw sequence files that were aligned to *Homo sapiens*; thus it acts as a high-end estimate of the time required to process data with XPRESSpipe. For a comparable dataset, cost to use an AWS (Amazon Web Services) computational node with similar specifications for the specified pipeline elapsed time in Table 2 would be approximately 25.76 USD using an Amazon EC2 On-Demand m5.8xlarge (however, significantly reduced rates are available if using Spot instances or by using the free tier) and storage cost would amount to around 17.41 USD/month for all input and output data on Amazon S3 storage (storage costs could be reduced as much of the intermediate data may not be relevent for certain users; however, raw input data should always be archived by the user).

## Availability and future directions

We have described the software suite, XPRESSyourself, an automated reference implementation of current best-practices in ribosome profiling data analysis built upon a synthesis of new

**Table 2. XPRESSpipe sub-module statistics for dataset GSE65778.**

| Process | Command | Wallclock Time | Max RAM |
|---|---|---|---|
| Curate STAR reference | `curateReference` | 00h 38m 34s | 34.03 GB |
| Truncate GTF | `modifyGTF -t` | 00h 03m 40s | 03.27 GB |
| Read Pre-processing | `trim` | 00h 08m 50s | 00.48 GB |
| Alignment / Post-processing | `align` | 07h 57m 44s | 38.03 GB |
| Read Quantification | `count -c htseq` | 03h 13m 04s | 00.16 GB |
| Isoform Abundance | `count -c cufflinks` | 00h 56m 44s | 02.36 GB |
| Differential Expression (n = 9) | `diffxpress` | 00h 07m 50s | 00.65 GB |
| Read Distributions | `readDistribution` | 00h 05m 00s | 00.28 GB |
| Metagene Analysis | `metagene` | 01h 45m 35s | 35.52 GB |
| Gene Coverage (n = 1) | `geneCoverage` | 01h 24m 00s | 19.32 GB |
| Periodicity | `periodicity` | 01h 08m 22s | 54.16 GB |
| Library Complexity | `complexity` | 01h 02m 57s | 01.52 GB |
| rRNA probe | `rrnaProbe` | 00h 00m 55s | 00.15 GB |
| Pipeline | `riboseq` | 16h 46m 19s | 54.16 GB |
| **Attribute** | | | **Value** |
| Total Raw Input | | | 257 GB |
| Total Output | | | 500 GB |
| Allocated Virtual CPUs | | | 32 |
| Minimum Allocated Memory | | | 62.50 GB |

The `geneCoverage` module was performed using a high-coverage gene. While some memory footprints are large in this test case, steps will scale based on available user resources. Input raw FASTQ files were uncompressed. The `metagene` and `geneCoverage` sub-modules used a conservative BAM file multiprocessing limit to avoid out-of-memory errors. XPRESSpipe v0.3.1 was used to generate these statistics.

tools, old tools, and pipelines. XPRESSyourself is perpetually open source and protected under the GPL-3.0 license. Updates to the software are version controlled and maintained on GitHub [52]. Jupyter notebooks and video walkthroughs are included within the README files at [52] (see Table 3). Documentation is hosted on readthedocs [88] at [50] and [51] (or archived versions can be found in S2 and S3 Files). Source code for associated analyses and figures for this manuscript can be accessed at [78]. The data used in this manuscript are available under the GEO (Gene Expression Omnibus) persistent identifier GSE65778 [89] for ribosome profiling data and under the dbGaP (Database of Genotypes and Phenotypes) Study Accession persistent identifier phs000178 [90] for the TCGA data used in the accompanying supplement (S1 File).

Although RNA-Seq technologies are quite advanced, standardized computational protocols are far less established for ribosome profiling. As we discussed in this manuscript, this becomes

**Table 3. Software description.**

| Project Name | XPRESSyourself |
|---|---|
| Project Home Page | https://github.com/XPRESSyourself |
| Archived Versions DOIs | 10.5281/zenodo.3338669, 10.5281/zenodo.3337897 |
| Operating Systems | macOS, Linux, centOS |
| Programming Languages | Python, R, Julia |
| Other Requirements | Anaconda |
| License | GNU General Public License v3.0 |

problematic when individuals or groups are not using best practices in analysis or may not be aware of particular biases or measures of quality control required to produce reliable, high-quality sequencing analyses. XPRESSpipe handles these issues through on-going curation of benchmarked software tools and by simplifying the required user input. It also outputs all necessary quality control metrics so that the user can quickly assess the reliability of their data and identify any systematic problems or technical biases that may compromise their analysis. Video walkthroughs, example scripts, and interactive command builders are available within this software suite to make these analyses accessible to experienced and inexperienced users alike. XPRESSyourself will enable individuals and labs to process and analyze their own data, which will result in quicker turnaround times of experiments and immediate financial savings.

One particular benefit of XPRESSyourself is that it consolidates, streamlines, and introduces many tools specific to ribosome profiling processing and analysis. This includes curating GTF files with 5′- and 3′- truncated CDS annotations, rRNA probe design for subtractive hybridization of abundant rRNA contaminants, automated quality-control analysis and summarization to report ribosome footprint periodicity, metagene coverage, and intron-less gene coverage profiles. These tools will help to democratize aspects of ribosome profiling analysis for which software have not been previously publicly available or difficult to access.

We demonstrated the utility of the XPRESSyourself toolkit by re-analyzing a publicly available ribosome profiling dataset. From this analysis, we identified putative translational regulatory targets of the integrated stress response (ISR) that may contribute to its neurodegenerative effects and their rescue by the small-molecule ISR inhibitor, ISRIB. This highlights the importance of re-analyzing published datasets with more current methods, as improved analysis methodologies and updated organism genome references may result in improved interpretations and hypotheses.

With the adoption of this flexible pipeline, the field of high-throughput sequencing, particularly ribosome profiling, can continue to standardize the processing protocol for associated sequence data and eliminate the variability that comes from the availability of a variety of software packages for various steps during sequence read processing. Additionally, XPRESSpipe consolidates various tools used by the ribosome profiling and RNA-Seq communities into a single, end-to-end pipeline. With these tools, genome reference formatting and curation are automated and accessible to the public. Adoption of this tool will allow scientists to process and access their data independently and quickly, guide them in understanding key considerations in processing their data, and standardize protocols for ribosome profiling and other RNA-Seq applications, thus increasing reproducibility in sequencing analyses.

## Ethics approval and consent to participate

Protected TCGA data were obtained through dbGaP project number 21674 and utilized according to the associated policies and guidelines.

## Supporting information

**S1 File. Supplemental text.** XPRESSpipe analysis of TCGA data, methods, and other supporting information.
(PDF)

**S2 File. XPRESSpipe documentation.** Archived version of the XPRESSpipe documentation. Current documentation can be found at https://xpresspipe.readthedocs.io/en/latest/.
(PDF)

**S3 File. XPRESSplot documentation.** Archived version of the XPRESSplot documentation. Current documentation can be found at https://xpressplot.readthedocs.io/en/latest/.
(PDF)

**S4 File. Interactive Plot 1.** Interactive plot file for XPRESSyourself vs Original TCGA processing of a sample with different gene types highlighted. XPRESSyourself-processed data used an un-modified Ensembl human build GRCh38v98 GTF file.
(HTML)

**S5 File. Interactive Plot 2.** Interactive plot file for XPRESSyourself vs Original TCGA processing of a sample with different gene types highlighted. XPRESSyourself-processed data used an un-modified Ensembl human build GRCh38v79 GTF file.
(HTML)

**S6 File. Interactive Plot 3.** Interactive plot file for XPRESSyourself vs Original TCGA processing of a sample with different gene types highlighted. XPRESSyourself-processed data used an un-modified Ensembl human build GRCh38v79 GTF file with no pseudogenes plotted.
(HTML)

**S1 Fig. Comparison between XPRESSyourself and other available software packages for ribosome profiling data analysis.** Black boxes indicate full functionality, blue boxes indicate partial functionality, grey boxes indicate incomplete or outdated functionality, and blank boxes indicate no functionality for the specified task. Rankings were compiled using the tools' documentation, manuscript, and codebase. If a function was not clearly described in any of these venues, a blank box was given.
(TIF)

**S2 Fig. Comparison between IGV browser and geneCoverage output.** A) Gene coverage from IGV (above) and XPRESSpipe (below) for SLC1A1. B) Gene coverage from IGV (above) and XPRESSpipe (below) for TSPAN33. Introns collapsed by XPRESSpipe. Green box, region displayed in corresponding IGV window.
(TIF)

**S3 Fig. Comparison between processed data produced by XPRESSpipe and original study.** Genes were eliminated from analysis if any RNA-Seq sample for that gene had fewer than 10 counts. A) Comparison of biological replicate read counts processed by XPRESSpipe. B) Comparison of read counts per gene between count data from the original study and the same raw data processed and quantified by XPRESSpipe. RPF, ribosome-protected fragments. Tm, tunicamycin. All $\rho$ values reported are Spearman correlation coefficients. XPRESSpipe-processed read alignments were quantified to *Homo sapiens* build CRCh38v98 using a protein-coding-only, truncated GTF.
(TIF)

**S4 Fig. Original ISRIB count data plotted against XPRESSpipe-processed data reveals systematic differences between the analytical regimes.** A) Selected highlighted genes show consistent differences between processing methods. B) Spearman correlation plots using the data table provided as supplementary data with the original ISRIB manuscript comparing biological replicates. RPF, ribosome-protected footprint. Tm, tunicamycin. All $\rho$ values reported are Spearman correlation coefficients.
(TIF)

**S5 Fig. Original ISRIB count data plotted against XPRESSpipe-processed data quantified using same reference version reveals mild improvement in comparability between the**

**analytical regimes.** Original samples were processed using Ensembl human build GRCh38v72, as in the original manuscript, and compared with the original count data provided with the manuscript. XPRESSpipe-prepared counts were thresholded similarly as the original data (each gene needed to have at least 10 counts across all mRNA samples). RepA, biological replicate A. RepB, biological replicate B. RPF, ribosome-protected footprint. Tm, tunicamycin. All $\rho$ values reported are Spearman correlation coefficients.
(TIF)

**S6 Fig. Cross-method analysis comparisons.** A) XPRESSpipe-processed data (orange) versus data as originally presented within original manuscript using original methods (green). B) Comparison of analyses using provided count table in original publication using DESeq2 (purple) versus original analysis provided in manuscript using DESeq1 (green). C) XPRESSpipe-processed (orange) versus originally-processed data (purple), both using DESeq2 for differential expression analysis. Light brown regions indicate overlap between gene lists. Thresholds used were the same as those used in the original study: $|\log_2(\text{Fold Change})| > 1$, FDR $< 0.1$.
(TIF)

**S7 Fig. Gene coverage plots for neurologically annotated genes passing strict thresholding.** Coverage plots were generated using XPRESSpipe's `geneCoverage` module, which collapses introns within the representation.
(TIF)

## Acknowledgments

The authors wish to thank Michael T. Howard for helpful discussions concerning ribosome profiling and sequencing analysis. The authors also wish to thank Mark E. Wadsworth, Ryan Miller, and Michael J. Cormier for helpful discussions on pipeline design. They also wish to thank T. Cameron Waller for helpful discussions related to pipeline design and biological analysis. The authors wish to thank members of the Rutter Lab for thoughtful discussions and comments. The support and resources from the Center for High-Performance Computing at the University of Utah are gratefully acknowledged. The results published here are in whole or part based upon data generated by the TCGA Research Network [62].

## Author Contributions

**Conceptualization:** Jordan A. Berg.

**Data curation:** Jordan A. Berg.

**Formal analysis:** Jordan A. Berg.

**Funding acquisition:** Jordan A. Berg, Jared Rutter.

**Investigation:** Jordan A. Berg.

**Methodology:** Jordan A. Berg, Jonathan R. Belyeu, Jeffrey T. Morgan, Jason Gertz.

**Project administration:** Jordan A. Berg.

**Resources:** Jordan A. Berg.

**Software:** Jordan A. Berg, Jonathan R. Belyeu.

**Supervision:** Aaron R. Quinlan, Jason Gertz, Jared Rutter.

**Validation:** Jordan A. Berg, Jeffrey T. Morgan, Yeyun Ouyang, Alex J. Bott.

**Visualization:** Jordan A. Berg.

**Writing – original draft:** Jordan A. Berg.

**Writing – review & editing:** Jordan A. Berg, Jonathan R. Belyeu, Jeffrey T. Morgan, Yeyun Ouyang, Alex J. Bott, Aaron R. Quinlan, Jason Gertz, Jared Rutter.

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
