## [Decision Letter · Decision Letter 0]

19 Sep 2019

Dear Dr Berg,

Thank you very much for submitting your manuscript 'XPRESSyourself: Enhancing, Standardizing, and Automating Ribosome Profiling Computational Analyses Yields Improved Insight into Data' for review by PLOS Computational Biology. Your manuscript has been fully evaluated by the PLOS Computational Biology editorial team and in this case also by independent peer reviewers. The reviewers appreciated the attention to an important problem, but raised some substantial concerns about the manuscript as it currently stands. While your manuscript cannot be accepted in its present form, we are willing to consider a revised version in which the issues raised by the reviewers have been adequately addressed. We cannot, of course, promise publication at that time.

Sincerely,

Aaron E. Darling

Software Editor

PLOS Computational Biology

[LINK]

Editor's specific comments:

Thank you for your patience while this manuscript was under review. As you will see the reviewers have highlighted some issues with the current manuscript that I think could be addressed in a major revision. While I am not concerned about the issues of the userbase requiring a GUI (it is ok to expect basic programming literacy among users), the manuscript could benefit from further articulating the advance over other software packages. As for the third reviewer's concerns regarding triviality of some of the functionality, one way you might address that is by making a case for comprehensive functionality available in a single well-engineered, tested, documented, "reference" implementation. There is a strong argument to be made against everybody having to (re)write their own functionality even for fairly simple tasks like sequence file parsing. I would encourage you to also describe more complex and advanced functionality as well. In your revisions please address the other concerns raised by the reviewers as well, of course.

Reviewer's Responses to Questions

**Comments to the Authors:**

Reviewer #1: I do not have extensive experience in the area of ribosomal profiling data analysis, but have reviewed the manuscript with respect to the bioinformatic tool description, analyses, and the accompanying software. The manuscript is carefully written, and it provides a useful pipeline for the uniform processing of ribosomal profiling data with best practices in mind. The software should be of utility to the community of bioinformatic analysts. The software is well documented at https://xpresspipe.readthedocs.io and https://xpressplot.readthedocs.io, and the GitHub repository for the paper provides all of the code used to produce the analyses in the paper.

I have the following minor comments:

* There are some ribosomal profiling software packages existing, although they probably handle some subset of the full pipeline. Michel [5] and Carja [8] are cited, but there appear to be at least a few more from searching with Google, e.g.,

https://doi.org/10.1186%2Fs12864-018-4912-6

https://doi.org/10.1016/j.cmpb.2018.10.018

And a few in the Bioconductor project are listed here

https://bioconductor.org/packages/release/BiocViews.html#___RiboSeq

I was expecting a table describing these, and e.g. a grid of checkboxes with respect to supported features compared to XPRESSpipe. Presumably this would also help to convince users to use XPRESSpipe as it would cover a broader range of the pipeline, and also indicate that some of these may now not be using best practices.

* The intron-removal for the purposes of visualization reminds me of this paper, which could be cited if appropriate:

https://genomebiology.biomedcentral.com/articles/10.1186/s13059-017-1284-1

* There appear to be a lot of noisy log2 fold changes in Figures 3A-D. Can these be fixed by filtering of low count features or use of pseudocounts in calculating the ratio, or the shrinkage techniques in the DESeq2 software that is used in calculating the TE?

* The software appears to be well described and documented, and the online documentation points to the GitHub Issues for user support. Consider also mentioning how users can obtain support in the manuscript. Additionally, the authors could provide information about how users would find out about changes to the software over time (e.g. as best practices evolve, where should users look to find descriptions of changes since the initial publication?).

Reviewer #2: The Manuscript describes XPRESSyourself, a comprehensive software package for analyzing the ribosome profiling data. As the Authors correctly point out, this experimental technique is gaining popularity, and at the same time, there is a lack of robust software tools to deal with idiosyncrasies if this data type. The XPRESSyourself tool appears to be expertly designed, with thoughtful considerations given to the accuracy, efficiency, robustness, and user-friendliness. The manuscript is well written and presents a biologically interesting story that aroused from the re-analysis of a previously published dataset. I have several suggestions that I believe could improve the paper.

1. Figure 1. It would be helpful to indicate directly in Figure 1 (in addition to the description in the body of the text) which tools and/or custom scripts are used for each of the steps, and which choices are available for the users. Also, specifying the format of the output files at each step (BED, BAM, txt, etc) would be useful for bioinformaticians who are willing to design their own pipeline extensions.

2. Figure 2. Presenting multiple scatter plots that all look pretty much the same seems to be wasteful of space. I would recommend presenting just one plot in (A) and (B) each, and moving all others to Supplementary materials. Instead, I would show the box-plots for the Spearman R between replicates and between methods (B). In addition to Spearman R, I would also recommend checking Pearson correlation of log(count+1), which sometimes gives a different picture.

3. In addition to correlation coefficients, it would instructive to compare the lists of significantly differentially expressed and TE genes yielded by the old and the current method. It can be presented as a Venn diagram. What are the genes with a significant TE increase in Tm treatment that were identified with XPRESSyourself but not with the old method?

4. I would recommend that the discussion about computational costs and requirements is moved to the main text, as it is important for many users. Also, it would be interesting to break the overall run-time and RAM requirements into separate values for each of the tools/modules.

5. “...systematic biological artifact…” If it’s biological, it’s not an artifact

Reviewer #3: The authors suggest a toolkit (XPRESSyourself) for better analyzing ribo-seq (which is mainly based on various previous software/tools ) and demonstrate it via the analysis of publicly available ribosome profiling data related to neurodegenerative phenotypes. As I describe below my impression is that the tool does not include novel enough aspects to be published in a journals such PLoS-CB. Nevertheless, I provide below some suggestions for improvements.

Major

------

1) They provide a code and not a GUI; thus a biologist with no programming experience can use it.

This means, that all the researchers that can use this code have programming expertise and thus can easily write all the simple analyses they provide or call the functions/tools that they use (since this is very easy to do I guess that this is what the potential users will typically prefer to do). Thus, eventually it is not clear to me who will use this tool.

2) There are many tools for analyzing ribo-seq (in addition to the one cited see e.g. PMID: 30049792, https://www.biorxiv.org/content/10.1101/106922v1.full, PMID: 27347386, PMID: 28158331, PMID: 30401579 .. this is a very partial list you should search for more additional tools). I do not find a clear explanation re. why we need an additional tool (you should clearly explain what are the *non trivial* aspects that you provide ?).

3) " For example, the pile-up of ribosomes at the 5'- and 3'- ..."

The authors give a long example re. why their tool is needed: it is the first to deal with the bias at the coding sequence ends. However their solution is trivial -- cutting/ignoring the ends of the coding sequence; any user that can use their code can *very* easily write a code for ignoring the ends.

I thought that they could give a novel/sophisticated solution that will deal with the fact that these regions do also include important signals (e.g. PMID: 25505165) that we may want to specifically study.. or suggest what is the exact region to remove to decrease bias but not removing important signals. Without such a solution their suggested feature is not helpful.

4) There are additional previous papers on ribo-seq biases/challenges that should be cited/mentioned (PMID: 27638886, PMID: 27160013 ).

5) The biological example they provide (in the section " Benchmarking Against Published Ribosome Profiling Data and New Insights ") is not very convincing: first, I guess that the improvement is mainly due to the usage of a new alignment tool (STAR); thus, this is mainly an advertisement to STAR. Second, most of the results are similar in the old and new analyses. Third, they suggest that they found relevant down regulated genes that the old paper did not find; however, the comparison is not performed in a systematic manner: you should compare all the genes you found and the old paper found and give some general quantitative measure (over *all* of them) that shows that overall your are better (e.g. has the old paper found relevant genes that you haven't found? do you find false positive -- genes that seem to be differentially translated but not seem to be relevant?).

6) One challenge in the field is dealing with various splicing isoform in the case of ribo-seq. Do you have a solution for this while considering *all* the relevant isoforms ?

7) Please increase the fonts in figure 2.

**Have all data underlying the figures and results presented in the manuscript been provided?**

Reviewer #1: Yes

Reviewer #2: Yes

Reviewer #3: None

PLOS authors have the option to publish the peer review history of their article (what does this mean?). If published, this will include your full peer review and any attached files.

Reviewer #1: Yes: Michael Love

Reviewer #2: No

Reviewer #3: No

---

## [Decision Letter · Decision Letter 1]

20 Dec 2019

Dear Dr Berg,

We are pleased to inform you that your manuscript 'XPRESSyourself: Enhancing, Standardizing, and Automating Ribosome Profiling Computational Analyses Yields Improved Insight into Data' has been provisionally accepted for publication in PLOS Computational Biology.

In the meantime, please log into Editorial Manager at https://www.editorialmanager.com/pcompbiol/, click the "Update My Information" link at the top of the page, and update your user information to ensure an efficient production and billing process.

One of the goals of PLOS is to make science accessible to educators and the public. PLOS staff issue occasional press releases and make early versions of PLOS Computational Biology articles available to science writers and journalists. PLOS staff also collaborate with Communication and Public Information Offices and would be happy to work with the relevant people at your institution or funding agency. If your institution or funding agency is interested in promoting your findings, please ask them to coordinate their releases with PLOS (contact ploscompbiol@plos.org).

Thank you again for supporting Open Access publishing. We look forward to publishing your paper in PLOS Computational Biology.

Sincerely,

Aaron E. Darling

Software Editor

PLOS Computational Biology

Reviewer's Responses to Questions

**Comments to the Authors:**

Reviewer #1: The authors have satisfied all of my previous comments. In particular, the table with current state and supported features of existing ribosomal profiling pipelines is a useful addition. The description of updates in the software documentation is also a useful addition to the help.

Reviewer #2: All my comments have been satisfactorily addressed.

Reviewer #3: acknowledge the work done by the author and believe that the tool should be published somewhere. However, I haven't been convinced that this tool should be published in PLoS-CB given the many other tools for ribo-seq analysis that have been published in the recent years and the relatively low level of novelty/improvement.

**Have all data underlying the figures and results presented in the manuscript been provided?**

Reviewer #1: Yes

Reviewer #2: Yes

Reviewer #3: Yes

PLOS authors have the option to publish the peer review history of their article (what does this mean?). If published, this will include your full peer review and any attached files.

Reviewer #1: No

Reviewer #2: No

Reviewer #3: No

---

## [Editor Report · Acceptance letter]

23 Jan 2020

PCOMPBIOL-D-19-01346R1 

XPRESSyourself: Enhancing, Standardizing, and Automating Ribosome Profiling Computational Analyses Yields Improved Insight into Data

Dear Dr Berg,

I am pleased to inform you that your manuscript has been formally accepted for publication in PLOS Computational Biology. Your manuscript is now with our production department and you will be notified of the publication date in due course.

With kind regards,

Laura Mallard
